# Perspectives Regarding the Intersections between STAT3 and Oxidative Metabolism in Cancer

**DOI:** 10.3390/cells9102202

**Published:** 2020-09-29

**Authors:** Kyung-Soo Chun, Jeong-Hoon Jang, Do-Hee Kim

**Affiliations:** 1College of Pharmacy, Keimyung University, Daegu 42601, Korea; chunks@kmu.ac.kr; 2Tumor Microenvironment Global Core Research Center, College of Pharmacy, Seoul National University, Seoul 08826, Korea; jhjang17@snu.ac.kr; 3Department of Chemistry, College of Convergence and Integrated Science, Kyonggi University, Suwon, Gyonggi-do 16277, Korea

**Keywords:** STAT3, cancer metabolism, post-translational modification, mitochondria, redox regulation, oxidative stress

## Abstract

Signal transducer and activator of transcription 3 (STAT3) functions as a major molecular switch that plays an important role in the communication between cytokines and kinases. In this role, it regulates the transcription of genes involved in various biochemical processes, such as proliferation, migration, and metabolism of cancer cells. STAT3 undergoes diverse post-translational modifications, such as the oxidation of cysteine by oxidative stress, the acetylation of lysine, or the phosphorylation of serine/threonine. In particular, the redox modulation of critical cysteine residues present in the DNA-binding domain of STAT3 inhibits its DNA-binding activity, resulting in the inactivation of STAT3-mediated gene expression. Accumulating evidence supports that STAT3 is a key protein that acts as a mediator of metabolism and mitochondrial activity. In this review, we focus on the post-translational modifications of STAT3 by oxidative stress and how the modification of STAT3 regulates cell metabolism, particularly in the metabolic pathways in cancer cells.

## 1. Introduction

The signal transducer and activator of transcription (STAT) consists of seven protein members: STAT1, STAT2, STAT3, STAT4, STAT5a, STAT5b, and STAT6. These proteins control many different biological functions, including cell differentiation, proliferation, apoptosis, and inflammation [1]. Three of these proteins—STAT1, STAT3, and STAT5—have been reported to be involved in cancer development. STAT1 is known to play a prominent role as an activator of the antitumor immune response, while STAT3 and STAT5 are primarily involved in promoting cancer progression [1]. In particular, STAT3 is the most frequently implicated protein in solid cancers [2], as it has been demonstrated in xenograft mouse models. In many types of cancer patients, excessive activity of STAT3 is associated with poor survival outcomes. [3,4]. STAT3 is involved in the tumorigenesis process by inducing the transcription of several genes that regulate survival, resistance to apoptosis, metastasis, and angiogenesis in tumor cells [5]. In addition, STAT3 regulates a large number of genes involved in immune response and tumor immune surveillance [6].

Although normal cells produce ATP and biosynthetic precursors through glycolytic and oxidative metabolism, cancer cells reprogram their metabolic processes to facilitate rapid, invasive, and metastatic growth. Altered metabolism induces the activation of oncogenes or loss of tumor suppressor genes in multiple signaling pathways, and confers competitive advantages to transformed cells [7]. It is recognized that most tumor cells switch their metabolism to upregulated glycolysis instead of oxidative phosphorylation in the presence of oxygen. This switch is called the Warburg effect, which occurs favorably in rapidly proliferating cells, such as cancer cells [8,9]. However, it was recently reported that the acidic microenvironment of solid tumors is associated with aggressive tumor phenotypes and favors oxidative phosphorylation instead of glycolysis [10,11]. Hypoxic conditions lead to enhanced glycolysis and suppress mitochondrial oxidative phosphorylation, whereas nutrient shortage because of rapid proliferation is associated with reduced glycolysis and restoration of mitochondrial oxidative phosphorylation [12]. In other words, the regulation of metabolic activity of cancer cells appears to be dependent on the condition of the tumor microenvironment. Therefore, both glycolysis and mitochondrial oxidative phosphorylation are potential candidates for therapeutic targeting.

In mitochondria, one of the inevitable by-products of oxidative phosphorylation is reactive oxygen species (ROS), and the aberrant expression of oxidative phosphorylation-related proteins leads to increased ROS formation. In cancer cells, the production of ROS is elevated as a consequence of increased metabolic activity and mitochondrial dysfunction. ROS are tumorigenic due to their ability to increase cell proliferation, survival, and cell migration, in addition to initiating tumor progression [13,14]. However, it is also known that ROS can function as an anti-tumorigenic agent, sinceit can induce cell senescence and cell death [15,16]. Therefore, cancer cells need to maintain an elaborate balance of antioxidant levels to survive against ROS. Moderate ROS levels can support the survival of cancer cells through the activation of signaling pathways, such as STAT3, which contribute to tumor growth. In recent years, the role of STAT3 in ROS formation and oxidative metabolic regulation has garnered increasing attention. Mitochondrial STAT3 phosphorylation enhances growth and invasion of murine 4T1 breast cancer cells by increasing complex I coupling system and reducing ROS production [17]. The canonical and non-canonical STAT3 signaling help to shift energy metabolism according to ROS production. STAT3 may act by interacting with electron transport chain (ETC) components, but the molecular mechanism that contributes to this process remains controversial. We suggest that STAT3 is one of the representative proteins that can shift the metabolic processes of cancer cells by exerting pro-oncogenic activity. In this review, we summarize the functions and regulatory mechanism of STAT3 in cancer, and further explain the relationship between STAT3 activity and cancer metabolism in terms of oxidation regulation.

## 2. Structure of STAT3

STAT3 comprises of several distinct functional domains: an amino-terminal domain, a coiled-coil domain, a DNA-binding domain, a linker domain, a Src homology 2 (SH2) domain, and a carboxy-terminal transactivation domain [18] (Figure 1). The N-terminal region of STAT3 consists of two functional domains, including an amino-terminal domain and a coiled-coil domain, which is responsible for allowing the tetramer formation through the interaction between STAT3 dimers to stabilize the protein-DNA complexes [19]. In addition, this region is essential for the noncanonical activation of STAT3 through the nucleocytoplasmic shuttling of its unphosphorylated form [20,21]. The SH2 domain contains important solvent-accessible sub-pockets that can bind with activated substrate proteins and stabilize STAT-STAT dimer interactions [22]. Reciprocal binding of the SH2 domain to the phosphorylated form on the tyrosine 705 residue (Tyr705) induces STAT3-STAT3 dimerization, which in turn leads to nuclear translocation, DNA-binding, and transcriptional target gene expression. The DNA-binding domain provides a binding interface between protein and DNA, and then dimerizes STAT3 according to the active stimulus. Finally, as explained earlier, dimerized STAT3 monomers can bind to specific DNA motifs and initiate transcription of STAT3 target genes [23].

Generally, both protein structure and protein-protein interactions are regulated by post-translational modifications (PTMs) including tyrosine and serine phosphorylation, acetylation, and methylation. Moreover, it has been suggested that the modification of cysteine thiols can largely affect the 3-dimensional structure of redox-sensitive proteins and their functions [24]. Cysteine modification, such as *S*-glutathionylation and alkylation of STAT3 at cysteine residues, was found to modulate STAT3 signaling [25,26]. In the next section, we describe how the activity of STAT3 is regulated by various PTMs.

## 3. Regulation of STAT3 Activity through Post-Translational Modifications

PTMs are important to induce signal transduction that transfers chemical groups, such as phosphate and acetyl groups, from one protein to another. The activation of oncogenes or the inactivation of tumor suppressor genes in cancer cells modulates various PTMs of effector proteins, providing a signal for sustained proliferation. Many types of PTMs are involved in the regulation of survival, cell cycle, and proliferation, leading to abnormally fast growth of cancer cells. In addition, altered tumor metabolism has been recognized as an emerging cancer hallmark and is associated with the Warburg effect or aerobic glycolysis, consisting of enhanced lactate production and increased glycolysis [27]. As previously mentioned, STAT3 is one of the most representative proteins involved in all of these phenotypes. Therefore, the STAT3 signaling pathway may contribute to the modulation of metabolism through diverse PTMs to provide a metabolic advantage to cancer cells, thereby promoting tumor cell proliferation and tumorigenesis. The following section summarizes STAT3-associated PTMs and intracellular signaling pathways.

### 3.1. Tyrosine Phosphorylation

STAT3 is a latent transcription factor that shuttles from the cytoplasm to the nucleus by sensing intracellular signaling transduction in response to cytokines and growth factors. As depicted in Figure 2, once cytokines or growth factors interact with corresponding receptors, STAT3 are recruited to activated kinases and become activated through phosphorylation of Tyr705 by upstream receptor-associated tyrosine kinases such as Janus kinases (JAK) and Src family kinases [28,29]. For example, the binding of interleukin-6 (IL-6) family cytokines to receptors triggers the formation of a complex with gp130, which recruits and activates JAK2. Activated JAK2 phosphorylates several tyrosine residues of co-receptor gp130 to create docking sites for STAT3 binding and phosphorylates STAT3 at the Tyr705 residue [28,29,30]. In addition, growth factors that act as STAT3 activators bind to their corresponding receptors, followed by the phosphorylation of the Tyr705 residue to recruit the latent cytoplasmic STAT3 [31]. Moreover, nonreceptor protein tyrosine kinases, such as Src and Abl, are involved in the activation of STAT3 [32]. Under physiological conditions, the STAT3 signaling pathway is precisely modulated by negative regulatory proteins such as SH2 domain–containing protein tyrosine phosphatase (SHP)-1, SHP-2, suppressor of cytokine signaling (SOCS), and protein inhibitor of activated STAT (PIAS) [33,34,35]. The SH2 domain of SOCS3 can bind to the JAK2, thereby preventing STAT3 binding to JAK and subsequently inhibiting its activation [36]. PIAS3 interferes with the binding of STAT3 to its specific DNA target sequence [37]. Interestingly, both SOCS3 and PIAS3 are target genes of STAT3 and provide a negative feedback loop that regulates STAT3 activity. The loss of negative STAT3 regulators is observed in multiple cancers, which promote STAT3 hyperactivity.

### 3.2. Serine Phosphorylation

The second phosphorylation residue of STAT3 is serine 727 (Ser727) in the C-terminal domain. It can be phosphorylated by various serine kinases, including the MAP kinase family (e.g., ERK, JNK, and p38), protein kinase C, and mTOR [38,39,40]. The Raf-MEK-ERK signaling pathway triggered by Ras oncogenes is responsible for the Ser727 phosphorylation of STAT3. The mitochondrial pool of STAT3 can support transformation of cells with activated ERK but not PI3K [41].

The Ser727 phosphorylation of STAT3 exerts various functions. In chronic lymphocytic leukemia, STAT3 was constitutively phosphorylated at Ser727 but not Tyr705 [42]. In addition, the phosphorylation of STAT3 at Ser727 and Tyr705 is known to play a different role in regulating the fate of mouse embryonic stem cells. More specifically, Tyr705 is required for self-renewal of these cells, whereas Ser727 is involved in proliferation and optimal pluripotency [43]. Interestingly, phosphorylated STAT3 at Ser727 predominantly exists in mitochondria and, thus, does not function as a transcription factor. Phosphorylated STAT3 that is translocated to the mitochondria regulates ETC activity [44]. Moreover, mitochondrial STAT3 contributes to the Ras-dependent malignant transformation of Barrett’s epithelial cells [45] and promotes the growth of breast cancer cells [17].

The actions of mitochondrial STAT3 in controlling respiration and Ras transformation are mediated by the phosphorylation state of Ser727. It has been proven that cells expressing STAT3, with a mitochondrial localization sequence, exhibit enhanced tumor growth and complex I activity in breast cancer cells [17]. We postulate that the translocation of phosphorylated STAT3 (Ser727) to the mitochondria could be an important signal initiator in cancer cell metabolism. In later sections of this review, we will highlight a critical serine phosphorylation site on STAT3 in relation to cancer metabolism.

### 3.3. Acetylation

Although phosphorylation is the most important post-translational mechanism to regulate STAT3 activity, many studies have investigated the characterization of STAT3 acetylation. There are debates about whether acetylated STAT3 functions as a positive or negative factor to regulate various aspects of STAT signaling. Upon cytokine and growth factor treatment, STAT3 can be acetylated on multiple lysine residues by cyclic adenosine monophosphate response element-binding protein (CBP)/p300 histone acetyltransferase. Acetylation on the lysine 685 (Lys685) residue of STAT3 enhances its dimerization and nuclear localization, leading to DNA-binding and transactivation activity [46,47]. In addition, acetylation of STAT3 at Lys685 can promote cell proliferation through cyclin D1 expression. It has also been shown that CBP/P300 complexes with cancer stem cell markers such as CD44 and STAT3 in the nucleus elicit STAT3 acetylation at Lys685, promoting subsequent STAT3 dimerization and cyclin D1 expression. Lee et al. reported that acetylated STAT3 leads to the recruitment of DNA methyltransferase 1 to the SHP-1 promoter region to silence its expression by DNA hypermethylation [48]. In addition, the deacetylation of STAT3 can be caused by NAD-dependent silent information regulator protein (SIRT) 1, which is subtly controlled by nutritional status [49]. SIRT1-mediated deacetylation leads to the suppression of STAT3 Tyr705 phosphorylation and is accompanied by the inhibition of transcriptional activity. Furthermore, the mitochondrial localization of serine-phosphorylated STAT3 was found to increase significantly in SIRT1-knockdown cells compared with wild-type cells [50]. As another stimulus, insulin induces the acetylation of the STAT3 lysine 87 residue to promote its mitochondrial translocation and function [51]. In cancer cells with a low Warburg effect, STAT3 is constitutively acetylated and undergoes steady-state translocation into the mitochondria, where it is constitutively involved in energy metabolism. The introduction of STAT3 into prostate cancer (PC-3) cells greatly decreased the Warburg effect, as reflected by reduced lactate levels and elevated glucose-to-fatty-acid metabolism [51].

### 3.4. Methylation

STAT3 in the nucleus undergoes methylation at lysine residues, which has a contrasting effect on its activity. In an IL-6-dependent response, activated STAT3 is methylated at Lys140 by the histone methyltransferase SET9 in the nucleus after Ser727 phosphorylation, and is followed by the methylation of STAT3, which has negative regulatory effects on the transcription of its target genes, including *SOCS3* [52]. By contrast, the methylation at Lys49 and Lys180 of STAT3 by EZH2, the lysine methyltransferase subunit of the polycomb repressive complex 2, increases STAT3 activation through Akt signaling, and the methylation of specific site is required for transcriptional activity in glioblastoma cancer cells [53].

### 3.5. S-Glutathionylation

The reversible redox modification of cysteine residues in redox-sensitive proteins, such as STAT3, NF-κB, and PTEN, can occur under oxidative conditions, and its activity can also be regulated [26,54,55]. *S*-glutathionylation, the most representative example of redox modification, can be triggered by intracellular glutathione and is accompanied by the inhibition of STAT3 transcriptional activity through the reversible oxidation of thiol groups present in cysteine residues. The addition of glutathione impairs JAK2-mediated STAT3 phosphorylation by disrupting the accessibility of tyrosine, which affects protein structure and function. Particularly, cysteine modification of STAT3 at Cys328 and Cys542, within the DNA-binding domain, impairs the phosphorylation of this protein [26]. In addition, C48, a selective STAT3 inhibitor, alkylates Cys468, present in the DNA-binding interface, then blocks the accumulation of activated STAT3 in various cancer cells [25]. In other words, STAT3 activation can be modulated by the glutathionylation or oxidation of diverse cysteine residues. Such modifications may be the basis for establishing crosstalk between STAT3 activity and cellular metabolic conditions [56,57,58,59].

## 4. Redox Balance of STAT3 in Cellular Survival

ROS are derived from electrons of partially reduced molecular oxygen that are generated and transformed in a variety of cellular processes. The most physiologically significant ROS are superoxide anion (O_2_^−^), hydroxyl radical (OH), and hydrogen peroxide (H_2_O_2_). ROS are generally produced during cellular metabolism and are important for various cellular functions. Redox homeostasis can be maintained by the balance between ROS production and its scavenging system. Its disruption may cause excessive oxidative stress, which in turn contributes to the pathogenesis of various diseases, including cancer, neurodegeneration, and aging. The formation of excessive ROS can induce the inactivation of various redox-sensitive proteins by mediating its oxidation, which can lead to activation of survival signaling pathways. A lack of ROS in the immune system can cause disease states that impair an individual’s ability to fight against foreign invaders [60]. Therefore, a proper balance of ROS, which can serve between useful functions and toxic effects, is crucial for survival and protection in various cell types [61].

STAT3 is one of the representative redox-sensitive transcription factors and is known to contain critical cysteine residues that can be sensitive to oxidation in the DNA-binding domain. ROS have been recognized to regulate the activity or stability of STAT3 by PTMs. Interestingly, the oxidation and glutathionylation of specific cysteine residues under conditions of excessive oxidative stress impair the DNA-binding and transcriptional activity of STAT3 [56,57,58,59], whereas mild ROS production, along with tyrosine phosphorylation of STAT3, promotes nuclear translocation of STAT3 [62,63,64,65]. ROS produced by nicotinamide adenine dinucleotide phosphate (NADPH) oxidase can activate JAK2 by inhibiting protein tyrosine phosphatases, which promotes pancreatic cancer cell survival [63]. This suggests that STAT3 activity is affected by the concentration of ROS. Conversely, there are reports that STAT3 contributes to the homeostasis of intracellular ROS. It was demonstrated that both nuclear and mitochondrial STAT3 leads to attenuated ROS formation and increased ROS scavenging activity [66,67,68]. Constitutive activation of STAT3 promotes glycolysis through HIF-1α transcriptional response under conditions of oxygen deprivation and decreased mitochondrial activity [67,69].

Deregulated redox homeostasis is a hallmark of cancer cells, and the resulting alterations in certain metabolic pathways is frequently found in tumors [70]. Reciprocal crosstalk between redox balance and cancer metabolism is particularly emphasized in glycolysis, glutaminolysis, fatty acid oxidation, and the pentose phosphate pathway [70,71,72]. Mitochondria are widely recognized as a source of ROS, but also serve a primary role in energy maintenance in animal cells. ROS have been considered to play a critical role in regulating metabolism and progression in cancer cells [73]. Therefore, mitochondria, the organelle that regulates the synthesis of various metabolites, such as ROS and ATP, must play a key role in cancer progression. The activation of oncogenes is associated with elevated mitochondrial ROS. Induction of K-Ras expression induces mitochondrial dysfunction and ROS production to promote cancer development [74]. Oncogenic K-Ras can regulate HIF-1α and HIF-2α to modulate mitochondrial metabolism and ROS production [75]. STAT3 phosphorylation occurs at the earliest stages of K-Ras-induced pancreatic tumorigenesis and is maintained in invasive carcinoma [76]. This is most likely that increased glucose utilization and glutamine metabolism are promoted by the oncogenic activation of *Myc* and *K-Ras* genes, which is enhanced by the mitochondrial-dependent biosynthesis of macromolecules through increased ATP levels and an accelerated tricarboxylic acid cycle [77,78]. In the next section, we explore the signaling pathways or molecules associated with STAT3 that link to metabolic processes occurring in mitochondria and induce ROS generation.

## 5. ROS Regulatory Molecules Involved in STAT3 Activity

The integration of exogenous and endogenous H_2_O_2_ levels, as well as cytokine signaling, affect the localization and activity of the different STAT3 types, which controls cell growth and survival. This may be particularly relevant in cancer cells, which often rely on the level of ROS to promote survival and metabolic adaptation [79]. Therefore, we aimed to summarize the molecular mechanisms underlying the regulation of the redox state of STAT3 according to the level of ROS (Figure 3).

### 5.1. Prx2

Peroxiredoxin 2 (Prx2), a thiol-dependent peroxidase, is one of the major H_2_O_2_ scavengers within cells. It is oxidized when exposed to H_2_O_2_ and provides an antioxidant function by cycling between its reduced and disulfide-bonded forms. Prx2 oxidized by exposure to H_2_O_2_ binds to STAT3, which prompts the formation of sulfide-linked STAT3 and reduces the transcriptional activity of STAT3 [59]. Indeed, the expression of the redox-insensitive cysteine mutant STAT3 stimulated the activity of this transcription factor, resulting in increased cell growth rates.

### 5.2. Trx1

Thioredoxin 1 (Trx1) reduces oxidized cellular components and activates certain transcription factors [80]. As mentioned before, oxidized STAT3 induces the formation of an unphosphorylated, disulfide-linked STAT3 dimer, which is transcriptionally inactivated. To restore its transcriptional activity, oxidized STAT3 dimers must be reduced by Trx1, which can restore STAT3 linkage through the disulfide bonds in reduced form [59]. Trx1 converts to an inactive form when oxidized in the processing of STAT3 to the reduced form. Trx1 can then be reactivated by TrxR1 using NADPH [80]. TrxR1 contributes to cancer survival and progression by maintaining STAT3 in the reduced state, thus allowing STAT3 to be phosphorylated at Tyr705 to guide dimerization and transcriptional activity. To combat the oxidative stress induced by the enhanced metabolic rate of cancer cells, the expression of Trx and glutathione is upregulated [81]. Busker et al. reported that potential inhibitors of STAT3-dependent luciferase activity in cells were shown to target TrxR1 directly. In cancer cells, the inhibition of TrxR1 expression results in increased oxidative stress and the accumulation of oxidized Prx2 and STAT3, which blocks STAT3-dependent transcription [82]. Therefore, it is possible to exploit the relationship between STAT3 and TrxR1 to generate novel anticancer therapies. Thus, this work is expected to contribute to the development of concomitant inhibitors for these proteins.

### 5.3. Apurinic/Apyrimidinic Endonuclease 1/Redox Factor 1 (APE1/Ref-1)

The elevated expression of apurinic/apyrimidinic endonuclease 1/redox factor 1 (APE1/Ref-1) is associated with poor survival in numerous types of cancer and could be a potential biomarker to predict prognosis in patients with solid tumors [83]. APE/Ref-1 is a substrate of Trx1 that exerts redox control of transcription factors, such as AP-1 and NF-κB, and stimulates DNA-binding activity via the reduction of disulfide cysteine residues present in the DNA-binding domain of each transcription factor [84,85]. This control explains how the expression of APE/Ref-1 protein protects cells against the various genotoxic effects of ROS, while the downregulation of APE/Ref-1 sensitizes cells for apoptosis [86]. Interestingly, STAT3 activity is inhibited by the knockdown of APE/Ref-1 but does not affect its phosphorylation or nuclear translocation in pancreatic cancer cells [87]. Dual blockade of APE/Ref-1 and STAT3 promotes marked tumor cell apoptosis, which results in significant inhibition of migration [87]. The complexes containing STAT3 and APE/Ref-1 bind to the promoter region of the vascular endothelial growth factor and increases its protein expression in pancreatic and prostate cancer [88]. In response to oxidative stress induced by tert-butyl hydroperoxide treatment in prostate cancer cells, *S*-glutathionylation of STAT3 prevents STAT3 phosphorylation at Tyr705 and inhibits the formation of STAT3 dimers, which precludes STAT3 canonical transcriptional activity [89]. These results suggest the interplay between STAT3 PTMs and APE/Ref-1 is required for redox sensing of STAT3 through the redox-sensitive Cys328 and Cys542.

## 6. Regulation of Metabolic Signaling Pathway through STAT3 in Cancer

The link between redox signaling and cellular metabolic circuits has become increasingly important in cancer biology, both as a benefit to cancer cell survival and to confer therapeutic resistance. As a master regulator of metabolic fluxes, mitochondria play a crucial role in tumor biology. STAT3 is known to regulate both genes involved in metabolism and mitochondrial activity [90]. There are also reports of key associations that occur between the STAT3 signaling pathway and cellular metabolism. In this section, we will explore metabolic signaling that is regulated by STAT3.

### 6.1. Relationship between STAT3 and HIF-1α under Hypoxic Conditions

Hypoxia, a condition in which cells are starved of oxygen, is a known hallmark of solid tumors that triggers stress responses and induces resistance to chemotherapy and radiotherapy. Hypoxia is rapidly generated in the inner mass of proliferating and expanding tumor tissues. Tumor cells tend to accelerate the rate of glucose metabolism to support their fuel in an environment of limiting oxygen concentrations. A metabolic switch toward aerobic glycolysis is believed to be required for rapid proliferation of cancer cells [91,92]. Hypoxia-inducible factor (HIF) 1 is a transcription factor that regulates *VEGF* gene expression in response to hypoxia [93]. The regulation of STAT3 in the glycolysis process is primarily mediated by several interactions between STAT3 and HIF-1α. This indicates that STAT3 transcriptionally upregulates mRNA and protein levels of HIF-1α [90] and attenuates mitochondrial activity by downregulating genes present in mitochondria [67]. Constitutively active STAT3 promotes the enhancement of aerobic glycolysis and proliferation, as demonstrated by the inhibition of STAT3 in an experimental tumor xenograft model that triggered a decreased glucose uptake [67]. This metabolic function strongly contributes to the upregulation of HIF-1α, leading to an increase in pyruvate kinase 2 (PKM2) levels and, in turn, supporting cell proliferation and survival [91,94]. In a positive feedback loop, PKM2 localizes to the nucleus in rapidly growing cancer cells, where it phosphorylates STAT3 to increase its transcriptional activity. Persistent activation of STAT3 is linked to tumor-associated angiogenesis, where STAT3 is capable of modulating the stability and activity of HIF-1α, resulting in enhanced VEGF expression. Hypoxia-induced activation of STAT3, transactivates the VEGF promoter and increases the expression of VEGF transcripts [95]. VEGF, a potent angiogenic factor, is upregulated in a variety of cancers and contributes to angiogenesis in tumor tissues. The level of VEGF correlates with the progression of malignancy. Stabilized HIF-1α forms an active complex with transcriptional coactivator p300 and phosphorylated STAT3 at the VEGF promoter, which stimulates its expression in epithelial ovarian cancer [96].

### 6.2. Role of Mitochondrial STAT3 in Cancer Metabolism

Whether STAT3 is involved in aerobic glycolysis in the cytoplasm or oxidative phosphorylation in the mitochondria largely depends on the PTMs present on STAT3. The activity of mitochondrial STAT3 in tumor tissues is suggested to be dependent on STAT3 phosphorylation at Ser727. By contrast, cells expressing oncogenes, such as Src and Myc, which induce the transcriptional activity of STAT3 through phosphorylation of Tyr705, will be converted to aerobic glycolysis [66,97]. Furthermore, STAT3 appears to function as a hub that integrates the various pro-survival and growth signals of energy and respiratory metabolism via both mitochondrial and nuclear activities [98]. In particular, STAT3 phosphorylation at Ser727 has emerged as a key regulator of metabolic processes. The Ser727 phosphorylation in STAT3 is positively related to tumor stage and size in estrogen receptor–negative breast cancer. In addition, this phosphorylation promotes breast cancer growth through regulation of the ETC in mitochondria [17]. The amount of ROS released from complex I is minimized by serine-phosphorylated mitoSTAT3. Thus, decreased production of ROS leads to decreased apoptosis and increased tumor cell growth [17]. Zhang et al. speculated that phosphorylated Ser727 in mitoSTAT3 favors optimal activity of complex I, which results in less production of ROS, decreased release of cytochrome c from the inner membrane and/or decreased apoptosis, and a net increase in cell growth [17]. The expression of GRIM-19 potentiates ROS production [99]. The phosphorylation of Ser727 facilitates the binding of mitoSTAT3 to GRIM-19, which dampens the ability of GRIM-19 to produce ROS and exerts a positive growth effect by mitoSTAT3. Therefore, these findings suggest that the phosphorylation of STAT3 at Ser727 plays a novel role in cancer progression, independent of the phosphorylation at Tyr705.

In addition to its positive role in cancer cell survival, mitochondrial STAT3 contributes to Ras-dependent cellular transformation by promoting ETC activity, particularly with respect to complexes II and V [66]. K-Ras-driven myeloid malignancy and pancreatic cancer development mediated by IL-6 signaling and dependent on myeloid malignancy and pancreatic cancer development were shown to be dependent on mitochondrial activity through serine phosphorylation of STAT3, which enhance ETC activity and ATP production [41,66,97,100]. It is known that the gene associated with retinoic-interferon-induced cell mortality 19 (*GRIM-19*), a component of complex I of the ETC, is capable of interacting with STAT3. This interaction is necessary for the import of STAT3 into the mitochondria [101]. Mitochondrial-targeted expression of mutant S727A STAT3 attenuates tumor growth and the metastatic ability of murine breast cancer (4T1) cells, which correlates with the reduction of complex I activity under hypoxia and leads to enhanced ROS production [17]. Another regulator, phosphatase SHP2, triggers enhanced mitochondrial metabolism and functions as a potential player via decreased phosphorylation of mitochondrial STAT3 [102]. Intriguingly, STAT3 localizes to the endoplasmic reticulum, where it modulates Ca^2+^ flux by interacting with the Ca^2+^ channel IP3R3. STAT3-mediated IP3R3 degradation in the endoplasmic reticulum contributes to enhanced cellular resistance to apoptotic stimuli, which is dependent on Ser727 phosphorylation but not Tyr705 phosphorylation [103]. However, an opposing report indicated that increased expression of IP3R3 stimulated cancer cell survival by promoting cellular metabolism [104]. Mice with colon cancer, driven by mutations of the gene encoding β-catenin, undergo a STAT3-dependent increase in aerobic glycolysis and suppression of ETC activity, which are mediated by the upregulated expression of HIF-1α and Myc [67,105]. The mechanism of how STAT3 regulates mitochondrial activity is not yet clear; however, specifically targeting mitochondrial activity is postulated to suppress tumor growth.

### 6.3. Role of Tyrosine Phosphorylated STAT3 in Cancer Metabolism

As previously explained, aberrant STAT3 activity drives the expression of genes that promote the cancer phenotype, including proliferation, resistance to apoptosis, metabolic changes, metastasis/angiogenesis, and immune system derangement [6,106]. The STAT3 Tyr705 phosphorylation plays a pivotal role in all of the aforementioned functions, especially the functions related to cancer metabolism. STAT3 promotes aerobic glycolysis by increasing glucose consumption and lactate production in oral squamous cell carcinoma cells, thus inducing the migration and invasion of cancer cells, as well as the epithelial-mesenchymal transition process [107]. Elevated expression of GRIM-19 leads to decreased STAT3 expression and increased oxygen consumption, followed by a reduction of aerobic glycolysis and cell proliferation [108]. In addition, STAT3 promotes aerobic glycolysis in hepatocellular carcinoma cells by increasing the levels of hexokinase 2, an isoform of hexokinase. This enzyme promotes irreversible processes in glycolysis, and helps mediate phosphorylation of glucose and the formation of ATP in mitochondria, which results in the increased growth, survival, and metastasis of cancer cells [109]. As shown in Figure 4, the mechanisms by which STAT3 can be constantly activated in the glycolysis process of cancer cells have been comprehensively proposed.

### 6.4. Role of STAT3 Acetylation in Cancer Metabolism

STAT3 acetylation by CBP undergoes mitochondrial translocation in response to serum or insulin stimulation. In mitochondria, STAT3 forms a complex with the pyruvate dehydrogenase complex E1 and subsequently accelerates the conversion of pyruvate to acetyl-CoA. In cancer cells with a low Warburg effect, STAT3 is constitutively acetylated and localized to the mitochondria, where it maintains ATP synthesis in energy metabolism [51]. It has been reported that doxorubicin generates oxidative stress and apoptosis in a number of cell types via mitochondrial redox cycling. In neonatal rat cardiac myocytes, p300 markedly increases in response to doxorubicin treatment. Autoacetylation-mediated stabilization of p300 results in acetylation, stabilization, and phosphorylation of STAT3, to provide a protective role against acute oxidative stress [110]. Few reports have investigated the effect of STAT3 acetylation on metabolism; however, this effect appears to be an important mechanism in cancer metabolism. Further investigation is required to fully clarify whether the PTM of STAT3 is a fate-determinant for the metabolic switch of its oncogenic functions.

## 7. Concluding Remarks

This review highlights the new findings about STAT3 in cancer metabolism. Energy metabolism plays a key role in the progression and growth of cancer via metabolic switches, such as increased aerobic glycolysis and reduced mitochondrial activity [92]. Metabolic switches in cancer are known to be associated with survival and resistance to cancer therapy. The results from several studies indicate that STAT3 functions as an integration center for various cancer cell signals through its nuclear and mitochondrial activities. In recent years, a pro-oncogenic role for STAT3 serine phosphorylation has been widely reported, and is also reported to be associated with its functions in the mitochondria [111]. However, the mechanism by which STAT3 carries out its activities within the mitochondria remains unknown and controversial. These conflicting results may be due to different tumor microenvironments; therefore, the role of STAT3 in aerobic glycolysis and ETC regulation warrants further investigation. Additionally, another report found that tyrosine phosphorylation of STAT3 participates in the modulation of mitochondrial ETC activity and in oxidative phosphorylation in response to interferon-β treatment [106]. We propose that the signaling of mitochondrial STAT3, which is not yet clear, is linked to the oxidation of STAT3 and requires further attention of researchers to evaluate links with other. Therefore, we conclude that STAT3 is a key signaling regulator of cancer metabolism.

## Figures and Tables

**Figure 1 cells-09-02202-f001:**
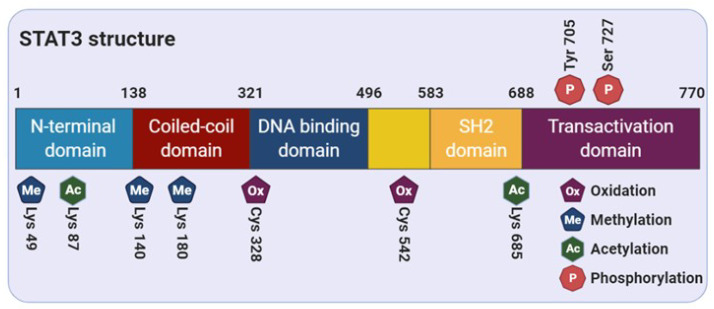
Functional domain of STAT3. The STAT3 protein consists of 770 amino acids and is divided into 5 distinct functional domains: the N-terminal domain, coiled-coil domain, DNA-binding domain, Src homology 2 (SH2) domain, and C-terminal transactivation domain.

**Figure 2 cells-09-02202-f002:**
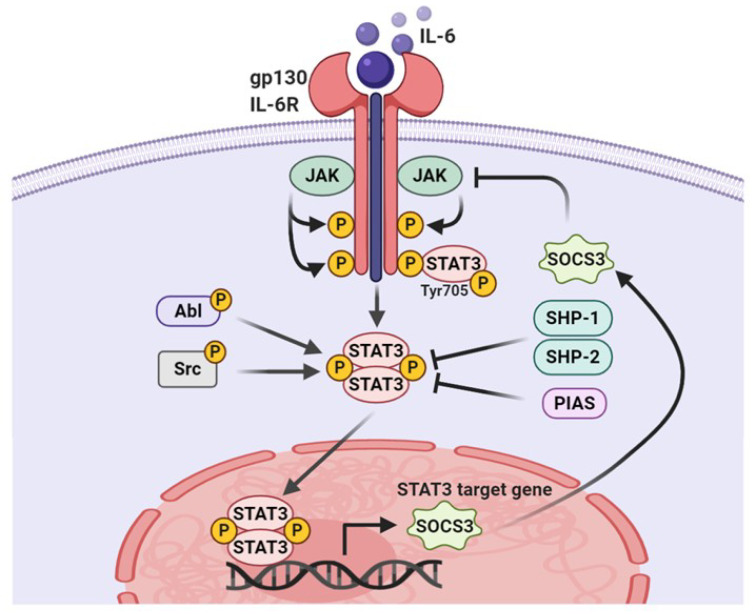
The canonical pathway of STAT3 signaling. IL-6 uses the JAK/STAT3 pathway to transduce its canonical signaling in cells. IL-6-induced STAT3 activation is transient, owing to the induction of the feedback inhibitor, SOCS3. Persistent activation of STAT3 by Src or Abl requires nucleocytoplasmic shuttling. SHP1, SHP2, and PIAS augment the negative regulation of STAT3 phosphorylation, which antagonizes survival signaling.

**Figure 3 cells-09-02202-f003:**
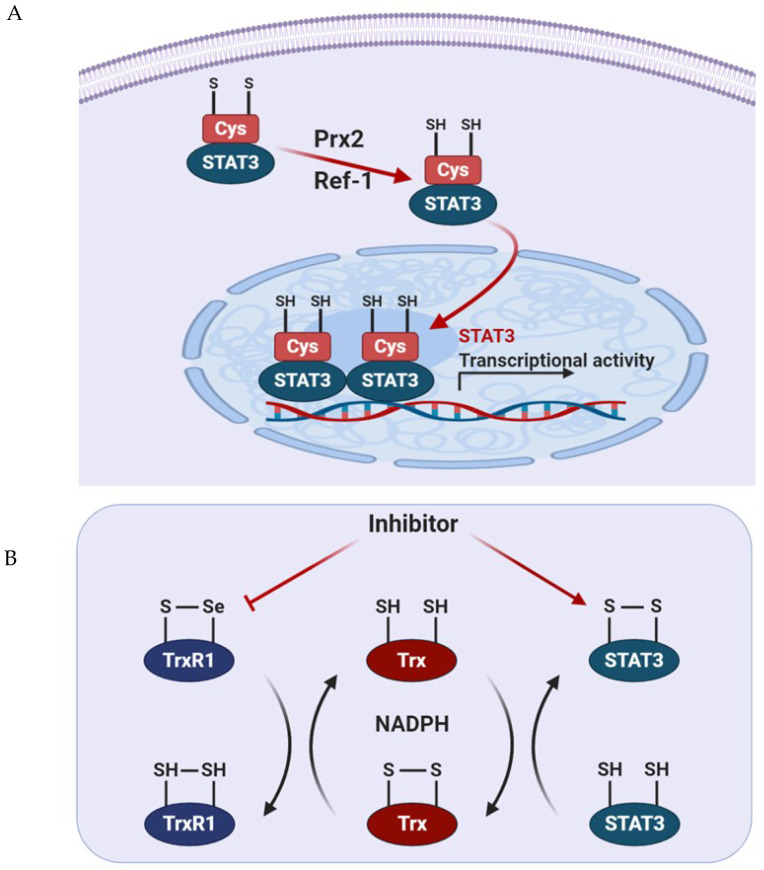
Intracellular signaling pathways modulating the redox status of STAT3. (**A**) Activation of Prx2 and Ref-1 results in the reduction of cysteine residues, thereby inducing transcription of STAT3. (**B**) Inhibition of TrxR1 results in the accumulation of oxidized STAT3, which blocks STAT3-dependent transcription.

**Figure 4 cells-09-02202-f004:**
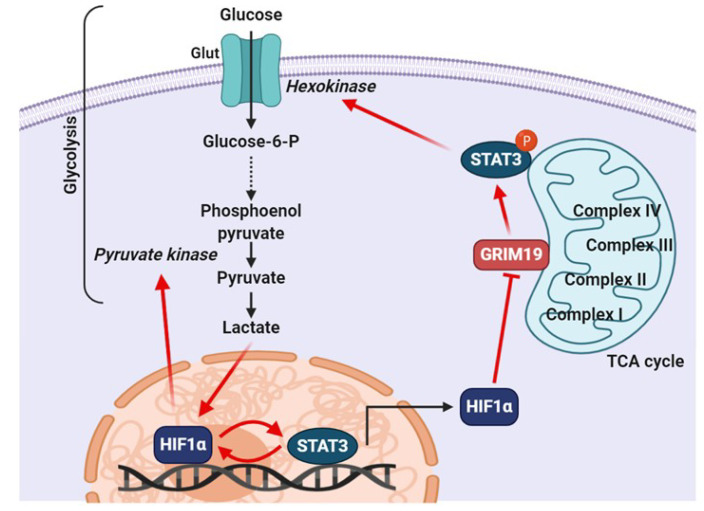
Proposed mechanism of constitutively active STAT3 in cancer cell glycolysis. Pyruvate produced via glycolysis is metabolized to lactate rather than to acetyl-CoA through lactic acid fermentation, and lactate can support the post-translational stabilization of HIF-1α in cancer cells. HIF-1α regulates PKM2 expression by binding to the hypoxia-response elements located within the *PKM2* gene. Upregulation of PKM2 expression results in the conversion of phosphoenolpyruvate into pyruvate. STAT3 works with HIF1α to activate HIF1α target genes and drive HIF1-dependent tumorigenesis under hypoxic conditions. Expression of HIF1α reverses the inhibitory effects of GRIM-19 on STAT3 phosphorylation. STAT3 could promote aerobic glycolysis of cancer cells by increasing the levels of hexokinase.

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
