# Peer review of "Perspectives Regarding the Intersections between STAT3 and Oxidative Metabolism in Cancer"

_cells, 2020, doi:10.3390/cells9102202_

Round 1
Reviewer 1 Report
Summary
This review captures a good amount of detail on STAT3, and the vast array of post-translational modifications that STAT3 (including phosphorylation, ubiquitination, methylation and S-glutathionylation status) is subjected to, which modulates its activity. I found sections 4 and 5 especially interesting, detailing well how reactive oxygen species (ROS) and redox levels can alter STAT3 activity. The importance of STAT3 in cancer and cancer metabolism is important to acknowledge, meaning that a review which collates this information is necessary. While a lot of information is captured in this review, it often feels like the links between the information is not clearly explained.
For example, the review suggests the roles and links between redox status and STAT3 in the context of cancer metabolism, however the links between sections are only described in a tenuous way. Sections 4 and 5 could be linked much more clearly to section 6. Another issue lies in the abstract, and what is set out as the aim of the review. The authors claim that the paper will focus on STAT3 activity and cancer metabolism, and how this promotes proliferation and metastasis. While the review mentions proliferation occasionally, a clear mechanism is never explained. For example, it would be good to detail how aerobic glycolysis aids in proliferation (i.e. by providing macromolecule supplies). Metastasis is mentioned minimally in the review. Either more detail must be inserted in the review about metastasis (and its links to metabolism), or it must not be mentioned as a focus. A good place to include metastasis may be in the section related to hypoxia and HIF-1α. For example: Hypoxia or STAT3 activation results in increased HIF-1α, which promotes angiogenesis, and the new disorganised vasculature leads to metastasis. Furthermore, STAT3 drives gene expression of ROCK within the Rho cell migration pathway, where ROCK/Rho are involved in cancer cell migration.
Throughout the review, there are significant grammar and formatting errors. These often detract from the clarity of the review. Additionally, figure legends are inconsistently formatted and poorly placed, giving a very rushed feel to the review in general.
Overall, the information which has been collected in this review is of relevance to the scientific community. However, more work still needs to be put in in order to clearly display the links between STAT3 and cancer metabolism. Further detail could be provided in some sections to provide a better picture of STAT3 signalling, and in regard to cancer metabolism. Severe issues in grammar and formatting need to be addressed as well, alongside issues with the figures and figure legends. Lastly, the review seems to lack a larger context. It may be beneficial to plainly state why it is necessary to understand metabolism in cancer and how STAT3 may be implicated in it (e.g. for new treatments by targeting STAT3 in certain cancer types). What clinical drugs are they that woul inhibit STAT3 for cancer therapies as an example, and there needs to be a review of clinical trials using STAT3 inhibitors.
Critical Analysis
- The abstract does not clearly state where research is currently lacking in this context. It could be good to mention the broader context of why we need to understand how STAT3 links to cancer metabolism. It is worth mentioning that STAT3 is a transcription factor when introducing it in the abstract.
- The abstract states that one of the things that the review will focus on is metastasis, but there is very little detail on the metastatic process, its relevance, and how STAT3 guides it throughout the rest of the review.
- The first 2 sentences of intro need referencing. I don’t think all that detail is encompassed in the first reference.
- In section 3, the authors provide a comprehensive description of the different post translational modifications (PTMs) which modulate the activity of STAT3, including referenced examples of each in different cancer types. However, given the text devoted to describing these PTMs, the review currently doesn’t adequately explain and discuss how metabolism effects/is affected by these PTMs within cancer.
- Improper use of grammar pervades the whole review and common errors include includes: incorrect use/omission of articles, wrong tense, incorrect use of the singular and plural etc. Furthermore within each paragraph sentence structure is often confusing and scientific terminology isn’t always used accurately or in a way that makes sense. These writing errors together adversely impact not only the coherence of the whole review, but also on the comprehension of many individual sentences and statements.
- In several instances, statements and sections of text are referenced with reviews, where it may be more appropriate to cite original primary research papers.
- Authors may want to consider including more discussion of STAT3 targets genes, especially those which promote metabolic transformation in cancer.
- Authors may wish to include a section on the development of STAT3 inhibitors in treating relevant cancers, with emphasis on those shown to normalise aberrant metabolism within cancer or those that exploit the metabolic perturbation resulting from STAT3 hyperactivity to selectively kill cancer cells.
- Line 28: The Statement “STAT3 is the most frequently implicated protein in solid cancers” needs clarification. I could not find that statement in the references provided for this piece of text. Points the authors may want to include here are that: Out of the STAT family of proteins, STAT3 is the most implicated in tumourigenesis. Whilst mutations in STAT3 are relatively infrequent in cancer, hyperactivation of STAT3 (as measured by Tyr705) is very common.
- Line 35-37: STAT3 also regulates a large number of genes involved in the immune response and tumour immune surveillance.
- Line 45-46: Apologies if incorrect, but I’m unsure if this sentence is entirely accurate. Within STAT3 the conserved N-terminal region isn’t directly involved in phosphorylated STAT3 dimer formation, but rather is thought to stabilise dimer interactions with weak STAT3 binding sites in DNA. The NTD has been shown to be part of the dimer interface within unphosphorylated STAT3 dimers. If this is what the authors mean, it needs to be made clear and referenced.
- Line 50-51: Unclear. Whilst SH2 domains do mediate dimerization of phosphorylated STAT3 dimers, Tyr705 is located within transactivation domain.
- Line 57: I would not define STAT3 as a ‘cytoplasmic transcription factor’, as STAT3 functions as a transcription factor in the nucleus. Maybe say that STAT3 senses intracellular signal transduction in the cytoplasm. Alternatively, could refer to STAT3 as a latent transcription factor that shuttles from the cytoplasm to the nucleus upon activation.
- Line 61-64: Good short summary of canonical STAT3 activation through IL6. However, would suggest extending sentence beginning line 63 to include that before activated JAK2 phosphorylates STAT3, JAK2 phosphorylates residues on co-receptor gp130 to create docking sites for STAT3 binding.
- Line 64-65: Within this sentence, how growth factor receptors phosphorylate STAT3 at Tyr705 needs clarifying. In the case of STAT3, upon ligand binding growth factors receptors phosphorylate STAT3 through their intrinsic receptor tyrosine kinase activity, in contrast to cytokine receptors which recruit JAKs to mediate kinase activity.
- Line 64-65: May benefit from including examples of growth factors which activate STAT3 through Tyr705 phosphorylation, e.g. EGF, PDGF and TGFα.
- Line 68-69: Clarification on SHP1 or SHP2. Specify what SHP stands for. (SH2 domain-containing protein tyrosine phosphatase.)
- Line 70: Perhaps specify SH2 domain of SOCS3 in JAK/STAT3 inhibition.
- Lines 70 – 72: Might be worth adding that SOCS3 and PIAS3 are STAT3 target gene and form part of a negative feedback loop that regulates STAT3 activity. Loss of negative regulators of STAT3 (through mutation, promoter hypermethylation and oncogenic miRNAs) is observed in multiple cancers, promoting STAT3 hyperactivity.
- Line 72: After this point the authors, for clarity, may consider adding a simple schematic figure showing STAT3 activation through canonical Il-6/JAK signalling to STAT3.
- Line 75 – 76: May benefit from noting here that many pathways dysregulated in different cancers signal through the serine kinases mentioned here (MAP kinases, protein kinase C and mTOR), e.g. Ras and PI3K signalling through ERK and mTOR. elaborating on the link between cancer and STAT3 activity.
- Line 76 – 77: The role of Ser727 phosphorylation is controversial/context dependent. Authors should elaborate on this point and include referenced examples of how Ser727 phosphorylation has been shown to be stimulatory and inhibitory to STAT3 activity.
- Line 81 – 82: Sentence needs clarifying, as it suggests to me Ser727 phosphorylation doesn’t affect STAT3s role as a transcription factor, where there are multiple studies that show it does.
- Line 89 – 90: Formatting error, meaning the title is at the top of the previous page.
- Line 94: “Modulates” seems imprecise to me, as this term could mean either enhance or reduce. Perhaps this wording should be changed to more precisely describe the ‘enhanced’ effect of Lys685 acetylation on STAT3 dimerization.
- Line 95: It may be worth adding here that acetylation of STAT3 at Lys685 can also promote cell proliferation through the cyclin CCND1. It has been shown CBP/p300 complexes with the cancer stem cell marker CD44 and STAT3 in the nucleus, eliciting STAT3 acetylation at Lys685 and promoting subsequent STAT3 dimerization and CCND1 expression.
- Line 95-97: It could be worth specifying which tumour suppressor genes are silenced (e.g., SHP-1).
- Line 95 – 97: It might be worthwhile stating that hypermethylation of tumour-suppressor genes by DNMT1 silences those genes.
- Line 99 – 100: Statement needs clarification. If known, does SIRT1 mediated deacetylation suppress both or either Tyr705 and Ser727 phosphorylation on STAT3.
- Line 102 – 103: Apologies if I’ve misunderstood, but I’m confused by the sentence beginning with “in contrast…”. In contrast to what? Insulin stimulated acetylation of STAT3 promoting translocation the mitochondria isn’t in contrast to the preceding point that knockdown of the deacetylase SIRT1 increases translocation of STAT3 to the mitochondria.
- Line 107: Reference #37 doesn’t mention methylation at Lys180 in STAT3. Reference #38 does and should be moved earlier in the text.
- Line 108 – 109: Sentence beginning line 108 needs clarification, STAT3 bound to specific promoters or all target gene promoters?
- Line 108 – 109: Potentially an over simplification of what authors of reference #37 found about methylation of STAT3 at Lys140. The paper also showed that for cells expressing mutants of STAT3 that can’t be phosphorylated at Lys140, induction of one group of genes was enhanced in response to IL-6 stimulation, but for other subsets of target genes induction wasn’t different or was repressed.
- Line 109: Authors should clarify the specific histone methyltransferase responsible for methylating STAT3 at Lys140. In This case SET9, as described in the reference for this portion of text.
- Line 115 – 116: This sentence needs clarifying, redox modification of cysteine residues where? Assuming this is STAT3.
- Line 116 – 118: What S-glutathionylation is needs to be better clarified here. Additionally, the section of the sentence which states “…reversible oxidation of thiol groups.” needs to be amended to clarify the thiol groups are on cysteine residues.
- Line 115 – 121: This sections needs properly referencing. I don’t understand the points that are raised in all four sentences and whether these points are fully encapsulated in reference #39.
- Line 120 – 121: According to the reference provided here (#39) and within figure 1, Cys542 is located within the linker domain, not the DNA binding domain.
- Line 126: First mentioning of ROS needs to say that it stands for reactive oxygen species.
- Line 127: Although examples of ROS are given in section of 5, it makes more sense to define ROS and provide examples here, were the review first introduces ROS.
- Line 131: It could be worth mentioning an example of the useful effects of ROS, i.e. immune system. This may give a little more context in to how ROS and STAT3 might be implicated in immune driven cancer.
- Line 135 – 138: Reference #46 demonstrates how ROS generated through NADPH oxidase can activate JAK2 by inhibiting JAK2s phosphatases. Maybe this should be its own referenced point in this section.
- Line 140 – 141: Might be worth emphasising phosphorylated nuclear and mitochondrial STAT3 decrease ROS production and briefly elaborating here how, e.g., HIF1A upregulation and increased glycolysis.
- Line 142-143: Might need a reference.
- Line 143: Specifically, also, elevated ROS levels are found in almost all tumours.
- Line 143: Which metabolic pathways are frequently altered in cancer? Some examples would be useful here.
- Line 142-152: This section is mostly derived from reference #52. To me, it feels a little bit jumbled and unclear in places. Perhaps consider rewording/restructuring, and also inserting more precise references (rather than referencing the whole review). Altogether, I think there is a shorter/simpler way of saying that mitochondria play a key role in cancer progression/metabolism.
- Line 145 – 146: Elaborate on “mitochondria are widely recognised as a source of ROS…”, how do mitochondria produce ROS, especially as it pertains to metabolism.
- There are a couple of instances where a large section of text is devoted to one reference of a review (For example, Ref 44, 52). It may be better to find the original references to cite, within that review, or from alternative sources/newer papers, for good practice.
- Line 158 – 161: Unsure what the authors are stating in this sentence?
- Line 161-162: Further detail should be given here, briefly state how ROS promotes survival and metabolic adaptation in cancer.
- Line 162: It could be relevant to mention that ROS also promotes cancer by directly damaging DNA.
- Line 164: Funny formatting here in the word spacing.
- Figure 1 should be closer to the relevant section (Maybe just after 3.5?). Also, figure legend is over the page. Also, “The” is bold in the figure legend text (format error). Slight pixilation if you zoom in on it. Could be better if saved in a different format, or re-made slightly larger, but not bad as is. Perhaps C-terminus should be labelled.
- Figure 1 is not referenced within the text at any point.
- Clarity of Figure 1 could be improved by including as part of the figure, the principal enzymes responsible for addition of post-translational modifications next to indicated residue position within figure. Alternatively, the authors could summarise this information in the figure legend.
- Figure 2: It feels like the figure legend could contain a little more detail on what is being represented here, possibly because of the location (I’m seeing it before having read what Trx does in redox balance). It might be worth having the figure a little lower (below the relevant sections which include an explanation of Prx2 and other Redox balancing molecules).
- Figure 2 could also be edited to more clearly show that the disulphide linking results in inactive STAT3 dimers. Also for consistency, either “Trx” or “Trx1” should be used throughout, rather than a mix.
- Figure 2 has a different formatting than figure 1, and also lacks a title.
- Figure 2: The location of this figure doesn’t make sense. The figure could be positioned below section 5.3 so that reader will then be familiar with Prx2, Trx1 and Ref-1.
- Line 174-175: I could be mistaken, but I think the acronym should be made within the text, rather than a subtitle? Not sure if that is true or not.
- Section 5.2 seems like it could be restructured to aid in clarity. For example, clarify that Trx1 converts to an inactive form when oxidized in the processing of STAT3 to the reduced form. Trx1 can then be re-activated by TrxR1, using NADPH.
- Line 188: Perhaps clarify that TrxR1 is essential for cancer cell survival by maintaining STAT3 in a reduced state. I.e. Therefore, TrxR1 contributes to cancer survival and progression by maintaining STAT3 in the reduced state, thus allowing STAT3 to be phosphorylated on tyr705 to guide dimerization and transcriptional activity.
- Line 190: Wording quite unclear. Makes it seem like the inhibitor is on the luciferase assay, rather than (what I assume) is an inhibitor of STAT3 or other targets.
- Line 195: Authors should mention Ref1/Ape1 that required for redox sensing of STAT3 through the redox sensitive Cys328 and Cys542 residues. These residues were previously mentioned, and is how Ref1 is absolutely required for STAT3 activation. This mechanistic link was omitted in this review.
- Line 195 – 205: Should be noted within section 5.3 that Ref-1 expression is upregulated in a number of human cancers and is correlated with poor patient survival.
- Line 195 – 205: Authors may want to consider and include that Ref-1 is itself a substrate for Trx1.
- Line 202 – 203: I think the information within this sentence is incorrect. Through its redox activity, Ref-1 acts to reduce specific cysteines in the DNA binding domain of STAT3, stimulating STAT3s DNA binding activity. Furthermore, reference #60 demonstrates how Ref-1 negatively regulates NRF2 (nuclear factor erythroid-related factor 2) and doesn’t look at STAT3 activity.
- Line 203 – 205: This sentence needs elaborating on, unsure on why its included currently. Authors should also stress that complexes containing STAT3 and Ref-1 binding to VEGF promoter and increasing its expression was discovered in pancreatic and prostate cancer.
- Line 202-203: It is not clear to me what this means exactly.
- Figure 3: Figure legend is very vague, and lacks a title (or has a title with no legend). Needs more of a description on what STAT3 is doing within glycolysis and metabolism.
- Line 214: For detail, it may be better to specify why hypoxia occurs (i.e. limited oxygen diffusion distance). Also, could be worth briefly mentioning the effects of hypoxia on other tumour characteristics (i.e. treatment resistance). It would also make sense to mention the role that both Hif-1α and STAT3 play in angiogenesis, since that is a primary response to hypoxia, and cancer cells need to establish suitable vasculature in order to grow past a certain size. While I appreciate this paper is mainly to do with metabolism, it seems lacking to not mention angiogenesis within this context, at least in passing.
- Line 218: Specify some genes which might be upregulated in response to hypoxia, and what they may do. E.g. VEGF.
- Line 221: Perhaps better to say STAT3 activity can downregulate transcription of mitochondrial genes rather than proteins.
- Line 226: Clarify that this is a positive feedback loop.
- Line 230: The word “prefers” seems odd to me. It might be better to use something like “promotes” instead.
- Line 234: Might be useful to specify examples of which oncogenes, to show which processes link to STAT3 activity, for a wider picture.
- Line 240 – 241: Clarify how Ser727 phosphorylation of STAT3 regulates the electron transport chain in mitochondria.
- Line 264: Line break needed to separate the concluding sentence from the previous section which talks about colon cancer. Perhaps shift the colon cancer section upwards a little bit, and link more to the other points. The summarising statement could do with a little more expansion and detail. I.e. “while the nuclear activity of STAT3 does (things to promote cancer), the mitochondrial activity of STAT3, while not yet clearly understood, is likely to suppress tumor activity by…”
- In section 7, it could be useful to briefly mention the broad roles that STAT3 plays in processes aside from metabolism, just to capture its diversity.
Examples of grammatical errors(there was too many this this manuscript to indicate all of them: I would recommend that additional editing/proof reading is carried out before resubmission)
- Review needs to be thoroughly proof read to tighten up the sentences and correct many spelling mistakes (typos).
- An example of this is in the first sentence of the abstract: ‘Signal transducer and activator of transcription 3 (STAT3) is the major molecular switch which plays an importnat role in the communication between cytokines and kinases, thereby regulating transcription of genes involved in a various biochemical processes such as proliferation, migration and metabolism of cancer cells.’
That would be better written as: ‘Signal transducer and activator of transcription 3 (STAT3) functions as a major molecular switch that plays an important role in the communication between cytokines and kinases, thereby regulating transcription of genes involved in various biochemical processes such as proliferation, migration and metabolism of cancer cells.’
- In an abstract, it would be best not to use terms like, etc.
- Line 26: bad grammar; ‘as an activation of the’ should be ‘activator’
- line 27: bad grammar; ‘primarily active in promoting cancer progression’ should be ‘primarily involved in’

- Line 32: Confusing link between two sentences that does not clearly explain what is occurring to the reader: ‘This switch is called the Warburg effect, which occurs favorably in rapidly proliferating cells such as cancer cells [4,5]. In other words, abnormally activated growth and survival signals that drive tumorigenesis facilitate the recovery of nutrients and increase the biosynthesis of cellular components, including the lipids and nucleotides [6].
As an example, this clearer second sentence would link in better to what was previously written: ‘Aerobic glycolysis increases the generation of biosynthetic lipid and nucleotide precursors needing for rapidly proliferating cells, which drives tumorigenesis.’
- Line 58: Poorly constructed sentence: ‘Once cytokines or growth factors interact with corresponding receptors, following to activation of receptor-associated tyrosine kinases such as Janus kinases (JAK) and Src family kinases.
- Line 70: ‘SOCS3 could bind to the JAK and the cytokine receptors’ should be ‘SOCS3 can bind to the JAK and the cytokine receptors’ and PIAS3 should be defined as a SUMO (small ubiquitin-like modifier)-E3 ligase.
- Line 76: ‘It is controversial to function of Ser727 phosphorylation in STAT3.’ needs to be rewritten to improve sentence.
Reviewer 2 Report
In this review article, the authors have provided an overview on the role of STAT3 in metabolism. The review is well written. But additional information could be included in Figure 2 and Figure 3, for example: how Prx2 and Ref-1 are activated and how TrxR is regulated in Figure 2; how ETC (please give whole name) is inhibited and what is the role of STAT3 in mitochondrial in Figure 3. Also the figure legends should include more description of the figures. More recent papers should also be included, such as the paper below.
Valle-Mendiola, A.; Soto-Cruz, I., Energy Metabolism in Cancer: The Roles of STAT3 and STAT5 in the Regulation of Metabolism-Related Genes. Cancers (Basel) 2020, 12 (1), 124.
Round 2
Reviewer 1 Report
The authors have dramatically improved the focus and quality of this paper on STAT3 and metabolism in cancer. The figures and legends also help with the narrative of the review, which now flows very well. I found this review a very interesting read, and predict that the readership of Cells will too.